# Body mass index trajectories and adiposity rebound during the first 6 years in Korean children: Based on the National Health Information Database, 2008–2015

Il Tae Hwang[1], Young-Su Ju[2], Hye Jin Lee [1], Young Suk Shim[1], Hwal Rim Jeong[3], Min Jae Kang [1] *

**1** Department of Pediatrics, Hallym University College of Medicine, Chuncheon, Korea, **2** Department of Occupational and Environmental Medicine, National Medical Center, Seoul, Korea, **3** Department of Pediatrics, Soonchunhyang University College of Medicine, Cheonan, Korea

* remoni80@gmail.com, mjkang@hallym.or.kr

## Abstract

### Purpose

We analyzed the nationwide longitudinal data to explore body mass index (BMI) growth trajectories and the time of adiposity rebound (AR).

### Methods

Personal data of 84,005 subjects born between 2008 and 2012 were obtained from infant health check-ups which were performed at 5, 11, 21, 33, 45, 57, and 69 months. BMI trajectories of each subject were made according to sex and the timing of AR, which was defined as the lowest BMI occurred. Subjects were divided according to birth weight and AR timing as follows: very low birth weight (VLBW), 0.5 kg $\leq$ Bwt $\leq$ 1.5 kg; low birth weight (LBW), 1.5 kg < Bwt $\leq$ 2.5 kg; non-LBW, 2.5 kg < Bwt $\leq$ 5.0 kg; very early AR, before 45 months; early AR, at 57 months; and moderate-to-late AR, not until 69 months.

### Main results

Median time point of minimum BMI was 45 months, and the prevalence rates of very early, early, and moderate-to-late AR were 63.0%, 16.6%, and 20.4%, respectively. BMI at the age of 57 months showed a strong correlation with AR timing after controlling for birth weight ($P < 0.001$). Sugar-sweetened beverage intake at 21 months ($P = 0.02$) and no-exercise habit at 57 months ($P < 0.001$) showed correlations with early AR. When VLBW and LBW subjects were analyzed, BMI at 57 months and breastfeeding at 11 months were correlated with rapid weight gain during the first 5 months (both $P < 0.001$).

### Conclusions

Based on this first longitudinal study, the majority of children showed AR before 57 months and the degree of obesity at the age of 57 months had a close correlation with early AR or rapid weight gain during infancy.

**Data Availability Statement:** Data cannot be shared publicly because of restriction. Data are available from the Korean National Health

Information Database (https://nhiss.nhis.or.kr) to limited researchers who meet the criteria for access to confidential data. Dataset names: Infant medical check-up cohort DB (https://nhiss.nhis.or.kr/bd/ab/bdaba022Yeng.do). The authors had no special access privileges others would not have.

**Funding:** This study was supported by Grant no. 2018-03 from the Kangdong Sacred Heart Hospital Fund.

**Competing interests:** The authors have declared that no competing interests exist.

## Introduction

The trajectory of an individual's body mass index (BMI) during life is quite variable. The BMI rapidly increases during the first year, then subsequently decreases and reaches a nadir around 4–8 years of age [1]. Thereafter, BMI increases again and this second rise following the last minimum BMI is referred to as the adiposity rebound (AR) [2]. The timing of AR is well-known to have a close relationship with obesity in later childhood, adolescence, and adulthood [3–7]. Although the definition of early AR varies according to studies, children who had early AR showed later obesity, insulin resistance, dyslipidemia, and hypertension [3, 5, 8]. Therefore, early AR is considered as a potent marker for obesity and metabolic syndrome.

The rapid postnatal weight gain showed an increased risk of early AR, obesity, and metabolic complications, especially in low birth weight (LBW) infants [4, 9, 10]. The definition of rapid weight gain during infancy is upward centile crossing in growth chart, which corresponds to changes in weight z-score over 0.67 [11]. Catch-up growth, which is attained by a period of supra-normal growth velocity, is essential to gain a normal growth in LBW infants, but an emphasis on adequate nutritional support during early infancy is important not to gain overgrowth too rapidly. And there is the positive link between rapid weight gain and later obesity in children born term or appropriate weight [12]. Therefore, higher risk of obesity of rapid grower is not confined only to preterm or small for gestational age infants, although rapid weight gain is most likely to occur in LBW infants. Like secular changes in body size and tempo of growth have occurred in most countries, the timing of AR has shifted by better nutrition, hygiene, and health status [13]. This shifting of AR timing was observed not only in children with overweight and obesity, but also in children with low BMI [14]. To date, there has not been any longitudinal data available for individual's BMI in Korea. However, the National Health Information Database (NHID) enabled us to explore sex-specific BMI growth trajectories of children born in the 21st century. Along with this, we also analyzed the timing of AR and related factors to early AR, including LBW children.

## Materials and methods

### Subjects and measurements

The longitudinal national cohort in NHID included the sampling unit (5% of total population) from infants and children born between January 2008 and December 2012 in South Korea. The NHID is a public database on health care utilization, health screening, socio-demographic variables, and mortality covering the whole population of South Korea and was formed by the National Health Insurance Service (NHIS) [15]. Health check-ups for infants conducted by the NHIS include mandatory examinations for growth, development assessments, dental exams, and infant care consultations reflecting health education. The infants and children may undergo seven screenings (Exams I–VII) at 4–6, 9–12, 18–24, 30–36, 42–48, 54–60, and 66–71 months of age. The median age at each exam (5, 11, 21, 33, 45, 57, and 69 months of age) was selected for analysis. The NHID covered data of 84,005 infants (based on Exam-I) and included 357,414 data points.

Data of height, weight, and BMI at each exam were retrieved. Most measurements were performed at primary health clinics. Length instead of height was used before 24 months of age and BMI was calculated as weight divided by (height or length)$^2$. As gestational age (GA) data were not available from the NHID, subjects with birth weight between 0.5 kg (which corresponds to the 3rd percentile for 23 weeks GA [16]) and 5.0 kg (which corresponds to +3.0 z-score of the male Korean standard) were included to remove outliers or errors. Height and weight at each exam was selected only when those values had increased compared to that in

| Initial number of data points = 357,414 | | | | | | | |
|---|---|---|---|---|---|---|---|
| Exam | I | II | III | IV | V | VI | VII |
| Exam age (month) | 4-6 | 9-12 | 18-24 | 30-36 | 42-48 | 54-60 | 66-71 |
| Median age (month) | 5 | 11 | 21 | 33 | 45 | 57 | 69 |
| N of total subjects | 67,301 | 65,591 | 63,522 | 63,674 | 47,730 | 31,357 | 18,239 |

Inclusion criteria

$0.5 \text{ kg} \leq$ birth Wt $\leq 5.0$ kg

Birth Wt < Wt-I < Wt-II < Wt-III < Wt-IV < Wt-V < Wt-VI < Wt-VII < +3 z-score of 69 months

25 cm < Ht-I < Ht-II < Ht-III < Ht-IV < Ht-V < Ht-VI < Ht-VII < +3 z-score of 69 months

At least 5 valid Ht, Wt, and BMI measurements during 7 times of health check-ups

| Final number of data points = 153,261 | | | | | | | |
|---|---|---|---|---|---|---|---|
| Exam | I | II | III | IV | V | VI | VII |
| N of total subjects | 27,143 | 23,983 | 24,662 | 24,867 | 24,804 | 17,590 | 10,212 |
| N of male | 13,841 | 12,226 | 12,554 | 12,721 | 12,631 | 8,944 | 5,191 |
| N of female | 13,302 | 11,757 | 12,108 | 12,146 | 12,173 | 8,646 | 5,021 |

**Fig 1. Flow chart of the study sample size.**

the previous exam, with minimum values of 25 cm (which corresponds to the 3rd percentile for 23 weeks GA [16]) and birth weight, respectively. An individual's longitudinal data were included when there were at least five valid height and weight measurements from seven health check-ups. Therefore, 27,143 infants (based on Exam-I) with a total of 153,261 data points were analyzed in this study (Fig 1).

Self-questionnaires were reported by a child's main parental care provider from Exams I to VII. Items of questionnaires were developmental screening (based on Korean Ages and Stages Questionnaires), visual acuity, and anticipatory guidance (accidents and poisoning, nutrition, prevention of sudden infant death syndrome, oral health, toilet training, socioemotional development, hygiene, and school readiness). Some questionnaires at each exam related to diet, exercise, lifestyle, and family history of metabolic syndrome were selected in this study. The Exam-II survey variables were breast feeding and the timing of weaning. The Exam-III survey variables were intake of sugar-sweetened beverage and use of additional salt to baby's food. The Exam-IV survey variables were intake of sugar-sweetened beverage and frequency of family meals with parents. The Exam-V survey variable was intake of sugar-sweetened beverage The Exam-VI survey variables were screen time in a day, exercise preference, and family history of metabolic syndrome. The Exam-VII survey variable was eating breakfast or not.

The institutional Review Boards of Hallym Medical Center (IRB# KANGDONG 2018-01-001) approved this study.

### Definition of subgroups according to birth weight, timing of AR, and rapid weight gain during early infancy

All subjects were divided into three subgroups according to birth weight as follows: very low birth weight (VLBW), $0.5 \text{ kg} \leq$ birth weight $\leq 1.5$ kg; low birth weight (LBW), $1.5 \text{ kg} <$ birth

weight $\leq$ 2.5 kg; and non-LBW, 2.5 kg $<$ birth weight $\leq$ 5.0 kg. There were 100 subjects (47 males and 53 females) in the VLBW group, 1,796 (802 males and 994 females) in the LBW group, and 25,247 (12,992 males and 12,255 females) in the non-LBW group.

Seven health check-ups did not provide sufficient frequencies to identify the exact age of AR. Therefore, we determined the timing of AR at which the lowest BMI occurred as in the previous study [5]. The definition of early AR also varied according to studies [2, 17]. In this study, we defined subgroups according to the timing of AR as follows: very early AR, timing of AR before 45 months of age (Exam-V); early AR, timing of AR at 57 months of age (Exam-VI); and moderate-to-late AR, no AR until 69 months of age (Exam-VII). If minimum BMI was not identifiable, we considered that individual's data as missing. Therefore, the number of subjects involved in AR timing analysis was 16,207 infants.

As rapid weight gain during early infancy is related to the development of later obesity [18], subjects were divided into two subgroups according to weight gain in the first 5 months (between birth and Exam-I). Because an individual's weight-for-age z-score data was not available in this study, we defined the rapid weight gain group who gained weight greater than or equal to 5.1 kg (which corresponds to the 0.67 changes of weight z-score for the male Korean standard who was born at median birth weight). While, the non-rapid weight gain group included subjects who gained weight $<$ 5.1 kg in the first 5 months.

The high BMI at age 5 years are also known as an important predictor to later obesity [13]. Therefore, we analyzed the BMI status at 57 months of age (Exam-VI) and classified as follows: underweight, BMI $<$ 5th percentile; normal weight, 5th percentile $\leq$ BMI $<$ 85th percentile; overweight, 85th percentile $\leq$ BMI $<$ 95th percentile; and obesity, BMI $\geq$ 95th percentile.

## Data analysis

Data were presented as mean ± standard deviation (SD). Comparisons of BMI trajectories over time between groups divided according to sex and AR timing were analyzed using a mixed model. A mixed effect linear model is used to compare nonindependent growth values within subjects over time between subgroups [19]. This model integrates two levels of observation (within and between-subjects) in a single model, therefore, it fulfills the needs of capturing the individual information over time [20]. Comparisons of BMI values at each exam point between males and females were analyzed by student's t-test. Comparisons of AR timing distributions between subgroups were analyzed using the chi-square or likelihood ratio test. Comparisons of questionnaire variables related to AR among subgroups based on AR timing and rapid weight gain during early infancy were analyzed using multiple logistic regression or analysis of covariance. The odds ratio for AR timing was calculated for BMI z-score at 57 months of age (Exam-VI) by multiple logistic regressions. All statistical analyses were performed in SAS Enterprise Guide 7.1 (SAS Inc., Cary, NC). *P*-value $<$ 0.05 was considered significant.

## Results

### BMI trajectories during the first 6 years and the timing of adiposity rebound

In all subjects, the mean BMI values at the ages of 5, 11, 21, 33, 45, 57, and 69 months were 17.8, 17.2, 16.3, 16.0, 15.9, 15.9, and 16.0 kg/m$^2$, respectively. There were differences in BMI values at each time point and BMI trajectories between male (Fig 2A) and female (Fig 2B) during the follow-up periods (all *P* $<$ 0.001). Of total subjects, 63.0%, 16.6%, and 20.4% showed very early, early, and moderate-to-late AR, respectively. The number of subjects were 10,209 (5,057 males and 5,152 females) in the very early AR group, 2,697 (1,390 males and 1,307

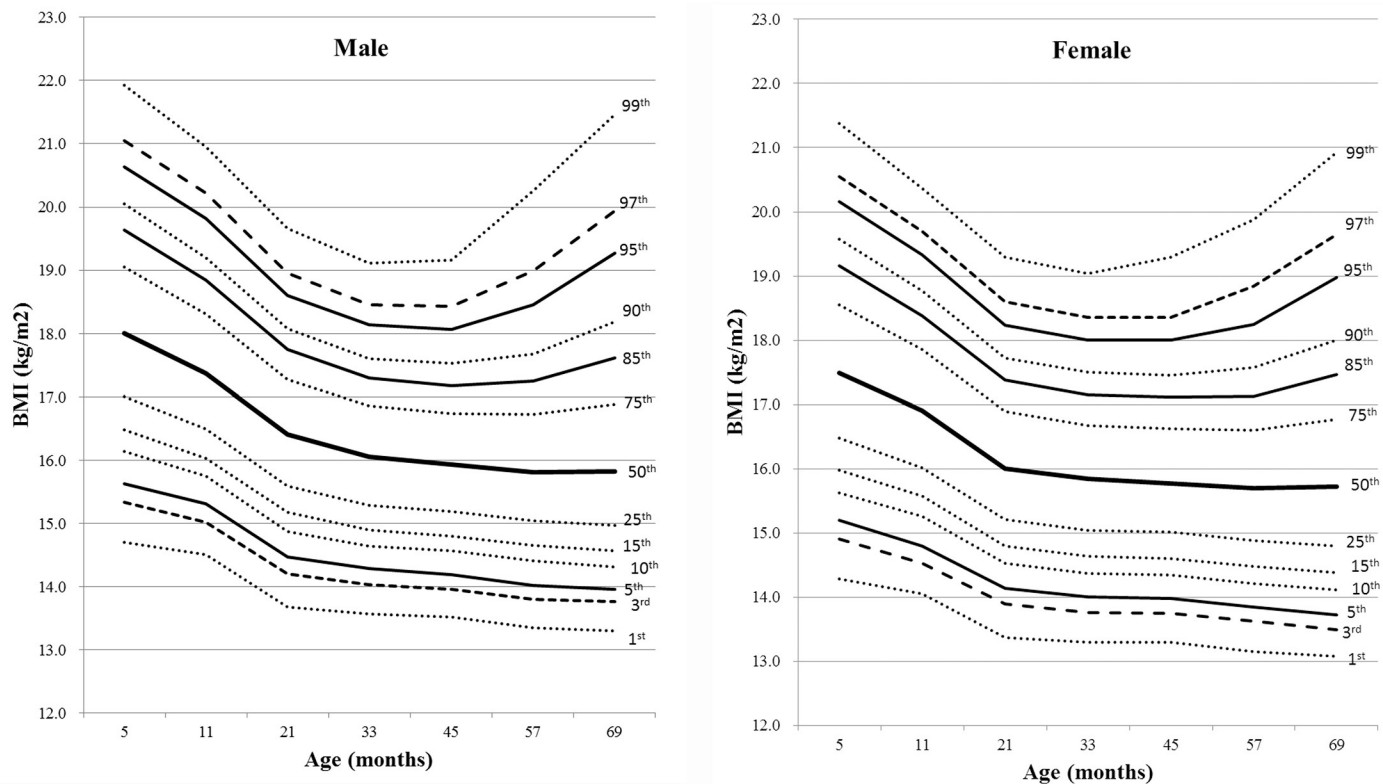

**Fig 2.** BMI percentile curves for subjects according to sex (A, males; B, females). Lines are 1st, 3rd, 5th, 10th, 15th, 25th, 50th, 75th, 85th, 90th, 95th, 97th, and 99th percentile curves from bottom to top, in order.

females) in the early AR group, and 3,301 (1,748 males and 1,553 females) in the moderate-to-late AR group. The median and of the timing of AR was 45 months (Exam-V) of age, in all subjects. Based on cumulative prevalence rates of the timing of AR, over 79% of subjects showed AR at age 57 months (Exam-VI) regardless of sex and birth weight (Table 1). BMI trajectories over time in the moderate-to-late AR group showed significant differences compared to those in the very early AR or early AR groups (all $P < 0.001$, Fig 3). However, there were no differences in BMI trajectories between the very early AR and early AR groups ($P = 0.89$).

## Related factors to the timing of adiposity rebound

BMI at age 57 months (Exam-VI) showed strong correlation with AR timing after controlling for birth weight ($P < 0.001$). The prevalence rates of overweight and obesity at age 57 months

**Table 1. The prevalence rates of the timing of adiposity rebound.**

| Exam | Median age | Total | Male | Female |
|------|-----------|-------|------|--------|
| I | 5 months | 0.2% | 0.3% | 0.2% |
| II | 11 months | 0.8% | 0.7% | 0.9% |
| III | 21 months | 10.4% | 9.2% | 11.6% |
| IV | 33 months | 28.5% | 28.7% | 28.3% |
| V | 45 months | 23.1% | 22.8% | 23.4% |
| VI | 57 months | 16.6% | 17.0% | 16.3% |
| VII | 69 months | 20.4% | 21.3% | 19.4% |

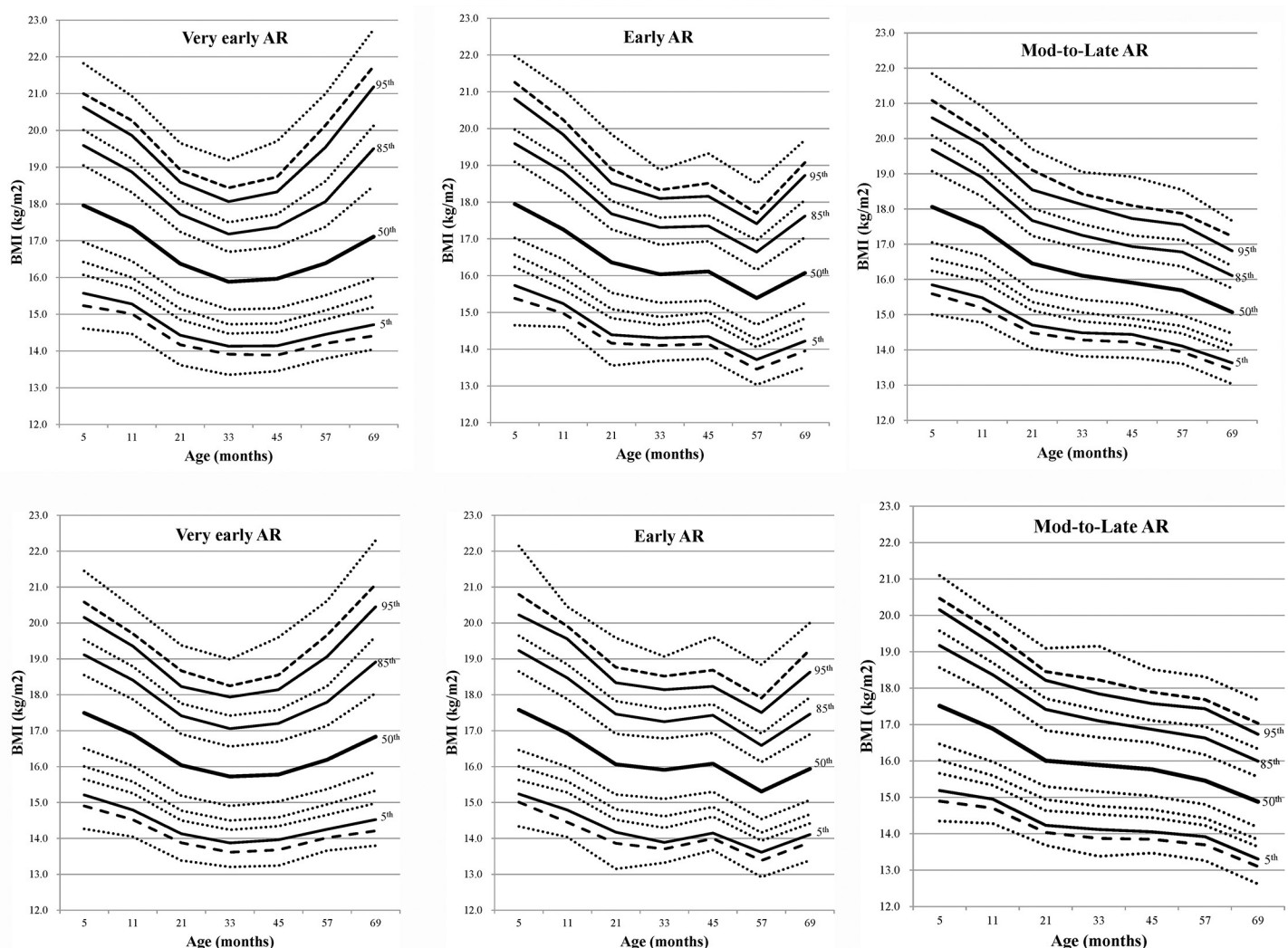

**Fig 3. BMI percentile curves for subjects according to timing of adiposity rebound.** (A-C) males and (D-F) females. Lines are 1st, 3rd, 5th, 10th, 15th, 25th, 50th, 75th, 85th, 90th, 95th, 97th, and 99th percentile curves from bottom to top, in order. Abbreviations: Mod-to-Late, moderate-to-late.

in the very early AR group (12.6% and 12.5%, respectively) were significantly higher than those in the early AR (4.9% and 2.1%, respectively) and moderate-to-late AR (5.9% and 1.4%, respectively) groups ($P < 0.001$). The obese children at age 57 months were 3.7 times more likely than normal weight children to have AR before 57 months (Table 2).

Based on questionnaires, weaning after 4 months of age (Exam-II, $P < 0.001$), sugar-sweetened beverage intake at 21 months (Exam-III, $P = 0.02$), and no-exercise habit at 57 months (Exam-VI, $P < 0.001$) demonstrated higher values in the very early AR group than in the early AR and moderate-to-late AR groups. However, breastfeeding at 11 months (Exam-II) showed no significant correlations with AR timing ($P = 0.20$), and screen time at 57 months (Exam-VI) showed contradictory results (Table 3).

## Rapid weight gain and adiposity rebound in low birth weight subjects

Greater weight gain during the first 11 months was observed in the VLBW group (mean 7.7 kg) than in the LBW (6.9 kg) and non-LBW (6.5 kg) groups ($P < 0.001$), and the increment

**Table 2. The distributions of BMI status at age 57 months by adiposity rebound groups and the odds ratios for adiposity rebound before 57 months.**

| BMI at age 57 months (n = 11,106) | Very early AR (n = 5,879) | Early AR (n = 2,697) | Mod-to-Late AR (n = 2,530) | AR before 57 months | |
|---|---|---|---|---|---|
| | | | | OR (95% C.I) | P-values* |
| Normal weight (n = 8,847) | 73.2% of VEAR | 84.9% of EAR | 89.1% of MLAR | Reference | |
| | 48.7% of NW | 25.9% of NW | 25.5% of NW | | |
| Underweight (n = 415) | 1.8% of VEAR | 8.2% of EAR | 3.6% of MLAR | 0.774 (0.670–0.895) | < 0.001 |
| | 24.8% of UW | 53.0% of UW | 22.2% of UW | | |
| Overweight (n = 1,018) | 12.6% of VEAR | 4.9% of EAR | 5.9% of MLAR | 1.756 (1.572–1.960) | 0.0009 |
| | 72.5% of OW | 12.9% of OW | 14.6% of OW | | |
| Obesity (n = 826) | 12.5% of VEAR | 2.1% of EAR | 1.4% of MLAR | 3.705 (3.201–4.288) | < 0.001 |
| | 88.7% of OB | 6.9% of OB | 4.4% of OB | | |

* by multiple logistic regression (adjustment for birth weight)

Abbreviations: Mod-to-Late, moderate-to-late; AR, adiposity rebound; OR, odds ratio; NW, normal weight; UW, underweight; OW, overweight; OB, obesity; VEAR, very early AR; EAR, early AR; MLAR, moderate-to-late AR

was prominent in the first 5 months (VLBW, 5.7 kg; LBW, 5.1 kg; and non-LBW, 4.8 kg). After 11 months (Exam-II), weight gain velocity did not differ between the VLBW and LBW groups.

**Table 3. Related factors to timing of adiposity rebound.**

| Variables | Age at survey | N | Category | Very early AR | Early AR | Mod-to-Late AR | P-values* |
|---|---|---|---|---|---|---|---|
| Breastfeeding | 11 months | 5,391 | Breast milk only | 66.3% | 15.7% | 18.0% | 0.20 |
| | | 8,345 | Formula, others | 65.8% | 15.1% | 19.1% | |
| Timing of weaning | 11 months | 693 | Before 4 months | 50.8% | 19.6% | 29.6% | < 0.001 |
| | | 13,150 | After 4 months | 66.7% | 15.1% | 18.2% | |
| Sugar-sweetened beverage | 21 months | 13,333 | < 200 cc/day | 65.1% | 15.7% | 19.2% | 0.02 |
| | | 978 | ≥ 200 cc/day | 69.5% | 14.0% | 16.5% | |
| | 33 months | 13,438 | < 200 cc/day | 65.3% | 15.7% | 19.0% | 0.71 |
| | | 913 | ≥ 200 cc/day | 65.6% | 14.8% | 19.6% | |
| | 45 months | 11,955 | < 200 cc/day | 61.4% | 18.0% | 20.6% | 0.71 |
| | | 822 | ≥ 200 cc/day | 62.5% | 18.0% | 19.5% | |
| Add salt to baby's food | 21 months | 12,081 | Yes | 65.5% | 15.6% | 18.9% | 0.69 |
| | | 2,304 | No | 64.6% | 15.8% | 19.6% | |
| Meal with parents | 33 months | 10,652 | ≥ 5 days/week | 64.8% | 16.0% | 19.2% | 0.12 |
| | | 3,730 | ≤ 4 days/week | 66.7% | 14.6% | 18.7% | |
| Having breakfast | 69 months | 8,099 | Yes | 29.1% | 31.9% | 39.0% | 0.98 |
| | | 361 | No | 29.4% | 32.1% | 38.5% | |
| Screen time | 57 months | 6,924 | ≤ 2 hours/day | 47.1% | 27.5% | 25.4% | < 0.001 |
| | | 2,458 | ≥ 3 hours/day | 41.0% | 30.0% | 29.0% | |
| Exercise habit | 57 months | 8,740 | Yes | 42.2% | 29.8% | 28.0% | < 0.001 |
| | | 639 | No | 90.6% | 5.5% | 3.9% | |
| Family history of metabolic syndrome | 57 months | 3,593 | Yes | 37.2% | 31.5% | 31.2% | 0.12 |
| | | 4,494 | No | 36.3% | 33.7% | 30.0% | |

*by multiple logistic regression (adjustment for birth weight)

Abbreviations: Mod-to-Late, moderate-to-late

Of the 1,129 VLBW and LBW subjects, 569 (50.4%) subjects were included in the rapid weight gain group in the first 5 months. There were no significant differences of timing of AR between the rapid and non-rapid weight gain groups ($P$ = 0.63). However, the rapid weight gain group showed greater obesity at age 57 months (Exam-VI, 5.8%) and less breastfeeding at 11 months (Exam-II, 20.4%) than the non-rapid weight gain group (obesity at age 57 months, 1.7%, $P < 0.001$; breastfeeding at 11 months, 28.1%, $P$ = 0.0003).

## Discussion

In this study, we demonstrated the BMI trajectories of infants and children born between 2008 and 2012 in Korea. In all subjects, the median time point of AR was 45 months of age, and 79.6% of subjects showed AR before 57 months. BMI at age 57 months along with diet and exercise habits had close relationships with early AR. Rapid weight gain during the first 5 months in the LBW and VLBW groups showed significant correlations with BMI at age 57 months and breastfeeding; however, AR timing was not relevant.

The timing of AR in the Korean growth standards, which was developed using cross-sectional data from the National Anthropometric Survey in 1997 and 2005 [21], was estimated as 5.5 years (66 months) in both sexes [13]. Considering that we had insufficient frequency of measured data in this study, the median timing of AR would be between 3.8 years (Exam-V) and 4.8 years (Exam-VI) of age, regardless of birth weight and sex. This is about one year earlier than that seen in the Korean standards, which may coincide with the global shifts in AR timing to an earlier age [14, 22–24]. The most recent survey conducted in Poland in 2010 showed AR timing of subjects (with median BMIs) as 5.2 years in boys and 3.0 years in girls, which was 2.3–4.4 years earlier than the values obtained in 1983 [22, 23].

Interestingly, BMI value at the age of AR was higher and BMI velocity after AR was greater in our subjects than those in the Korean standards [21]. In boys, BMI values at the age of AR for our subjects in the 50[th] percentile and in the corresponding Korean standard were 16.0 and 15.7 kg/m$^2$, respectively. And for those in the 95[th] percentile and in the corresponding Korean standard were 18.1 and 17.8 kg/m$^2$, respectively. BMI changes between 45 months of age (Exam-V, median AR timing) and 69 months of age (Exam-VII) were greater in male subjects in the 95[th] percentile (1.2 kg/m$^2$) than in the corresponding Korean standard (0.8 kg/m$^2$). Differences in minimum BMI values and BMI velocity after AR between our female subjects and the corresponding Korean standards showed similar results to that seen in males. Altogether, these results (AR before 4.8 years of age, higher minimum BMI values, and greater BMI velocity after AR) are in line with increasing obesity, which was also seen in the Fels Longitudinal study [25]. If children have similar genetic background, current environmental issues (such as decreased physical activity, increased screen time, inappropriate feeding practices, and abundant fast foods) compared to that in the past [26], may be held responsible for these results.

Currently, beyond the role of predicting obesity, earlier AR over time may imply the acceleration of growth and development compared to the past. Based on a longitudinal study, shifting of AR timing was observed not only for children with normal weight, overweight, and obesity, but also in underweight children [14, 22, 23]. Unfortunately, minimum BMI in underweight subjects of the 5[th] percentile was not confirmed during this period, because follow-up period (69 months) was short to confirm AR timing of all subjects in this study. However, Luo *et al.* also reported that earlier AR timing was observed in heavier and taller children compared to lighter and shorter children [27].

A positive relationship between small size at birth and early AR is well known [28]. Around 80% of children born small for gestational age show catch-up growth during the first year of life [29]. Accelerated growth of the entire body contributes to the accelerated growth of fat

cells, and this is a critical period for later body composition and fat deposition [1]. In this study, contrary to our expectation, AR timing showed no differences between the birth weight groups. There may be several reasons. First, regarding body compositions, thin children are predisposed to a smaller lean mass [30], which may rebound later than children with proper lean mass. Second, the number of subjects in the LBW and VLBW groups was small, relative to those in the non-LBW group in this study; therefore, this may attenuate the differences in AR timing. Third, gestational age is also an important factor for catch-up growth with a boundary of 37 weeks gestational age [17], but data of gestational period was unavailable in this study.

Gardner *et al.* suggested that by age 5 years, childhood obesity appears relatively resistant to change [31]. In this study, BMI at age 57 months (around 5 years) had a strong relationship with AR timing, and BMI trajectories between the very early AR (cutoff 45 months) and early AR (cutoff 57 months) groups did not differ significantly. Therefore, taken together, these results support the importance of obesity at age 5 years [32, 33]. Moreover, BMI at age 57 months was linked to the rapid weight gain group especially to the LBW and VLBW groups, in this study. This confirms existing evidences that weight gain during infancy is a predictor of later obesity [34]. Botton *et al.* reported that regardless of birth weight, rapid weight gain even in the first 6 months is associated with obesity [35]. Breastfeeding, the only factor known to have protective effect against obesity [13], showed negative correlation with early weight gain during the first 11 months in our study. Therefore, the benefits of breastfeeding in LBW and VLBW infants cannot be overestimated [8].

Among other factors related to early AR, parental BMIs, especially BMI of the mother, who mostly share an environment of diet habit or activity, was closely associated [25, 32, 36, 37]. However, the relationship between early AR and screen time [38], protein intake [39], or socioeconomic status [24, 32, 37] remains uncertain. Factors related to early AR such as later weaning, additional sugar-sweetened beverage intakes, and no-exercise habit showed expected results in this study, but as there are many confounding factors within the survey data, verifications are required.

A major limitation in this study is that the exact timing of AR is difficult to judge. The best way is to trace an individual's adiposity plot through visual inspection [2, 40]; however, most often, there is insufficient frequency of measured data to allow the use of this method. To compensate for this problem, various statistical approaches [5, 7, 40–42] or the researchers' own definitions [17, 32, 43] were used. Second, definitions of early AR also varied according to literatures. Generally early AR was defined before 5 years of age [2], however, over 75% of subjects had AR before 57 months in this study, so we subdivide into early AR (before 57 months of age) and very early AR (before 45 months of age). Therefore, while comparing AR timing between contemporary studies, methods of determining AR and definition of early AR should be carefully considered. We selected the lowest BMI as the timing of AR, therefore, median age of AR in this study might be lower than the exact timing of AR. Third, the small sample size at age 69 months (Exam-VII), no data after age of 6 years (unable to confirm late AR), and no data of gestational age could be limitations. The NHID only included 5% of total population and after deleting outliers, about 30% of initial subjects were finally included. However, this is the first national cohort study on infancy and early childhood based on individual longitudinal data; this may be sufficient to overcome any drawbacks of this study.

One of the critical periods in life for the development of obesity is the period of AR [44]. Most of Korean children in the 21$^{st}$ century showed early AR before 57 months and their diet and exercise habits may be related to early AR or early weight gain during infancy. This may be a warning sign for later obesity or early maturation problems, such as precocious puberty. Therefore, our results support the need of population-based preventive interventions.

## Acknowledgments

We thank to the National Health Insurance Service providing this valuable data.

## Author Contributions

**Conceptualization:** Min Jae Kang.

**Data curation:** Min Jae Kang.

**Formal analysis:** Young-Su Ju, Min Jae Kang.

**Funding acquisition:** Il Tae Hwang, Min Jae Kang.

**Investigation:** Il Tae Hwang, Young-Su Ju, Min Jae Kang.

**Methodology:** Il Tae Hwang, Young-Su Ju, Hye Jin Lee, Young Suk Shim, Hwal Rim Jeong, Min Jae Kang.

**Project administration:** Min Jae Kang.

**Resources:** Il Tae Hwang, Young-Su Ju, Min Jae Kang.

**Software:** Young-Su Ju.

**Supervision:** Il Tae Hwang, Young-Su Ju, Hye Jin Lee, Young Suk Shim, Hwal Rim Jeong, Min Jae Kang.

**Validation:** Il Tae Hwang, Young-Su Ju, Hye Jin Lee, Young Suk Shim, Hwal Rim Jeong, Min Jae Kang.

**Visualization:** Il Tae Hwang, Min Jae Kang.

**Writing – original draft:** Min Jae Kang.

**Writing – review & editing:** Il Tae Hwang, Young-Su Ju, Hye Jin Lee, Young Suk Shim, Hwal Rim Jeong.

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
