## [Decision Letter · Decision Letter 0]

29 Jul 2020

PONE-D-20-11438

Body mass index trajectories and adiposity rebound during the first 6 years in Korean children: based on the National Health Information Database, 2008-2015

PLOS ONE

Dear Dr. Kang,

Thank you for submitting your manuscript to PLOS ONE. After careful consideration, we feel that it has merit but does not fully meet PLOS ONE’s publication criteria as it currently stands. Therefore, we invite you to submit a revised version of the manuscript that addresses the points raised during the review process.

Please make sure you take into account all of the comments and suggestions raised by the reviewer.

We look forward to receiving your revised manuscript.

Kind regards,

Jean-Philippe Chaput, Ph.D.

Academic Editor

PLOS ONE

Journal Requirements:

2.We note that you have indicated that data from this study are available upon request. PLOS only allows data to be available upon request if there are legal or ethical restrictions on sharing data publicly. For information on unacceptable data access restrictions, please see http://journals.plos.org/plosone/s/data-availability#loc-unacceptable-data-access-restrictions.

Additional Editor Comments (if provided):

Dear Author,

Your manuscript has been assessed by an expert in the field and myself. The reviewer has identified concerns with your paper that deserve attention. If you are ready to address all comments and suggestions raised, I will welcome a revised version of the manuscript. However, it does not guarantee acceptance.

Reviewers' comments:

Reviewer's Responses to Questions

**Comments to the Author**

1. Is the manuscript technically sound, and do the data support the conclusions?

Reviewer #1: Partly

2. Has the statistical analysis been performed appropriately and rigorously? 

Reviewer #1: No

3. Have the authors made all data underlying the findings in their manuscript fully available?

Reviewer #1: No

4. Is the manuscript presented in an intelligible fashion and written in standard English?

Reviewer #1: No

5. Review Comments to the Author

Reviewer #1: This is an interesting paper on a timely topic regarding the timing of the adiposity rebound in a large cohort of South Korean aged 0-6 years, but the take-home message is relatively confused. Here are my suggestions:

1/ the purpose/objective of this paper should be refined. There is too much information for a single paper. As you do not have information regarding gestational age of the participants, as the subgroups according to birthweight (VLBW, LBW and non-LBW) are really unbalanced, and as the timing of AR was not different between these subgroups, I suggest you delete them from the paper and only focus on the different types of rebounds, RWG and the factors associating with.

2/ I suggest you revise the name of your AR groups. What you call “Early AR” (AR before the age of 45 months (3.75 years)) actually meets the criteria of a very early adiposity rebound, and what you call “moderate AR” (AR at 57 months (4.75 years)) actually is the early AR. Even though you justified this point in the discussion section (“Generally, EAR was defined before 5 years of age, however over 75% of subjects had AR before 57 months…”), the data collection was not performed after the age of 6 years, so children exhibiting a late rebound were not considered in the analyses, and, consequently, median age of AR might appear lower than it is. In a previous paper (https://doi.org/10.3345/kjp.2018.07227) you reported that in both boys and girls in the 50th percentile of the curve, adiposity rebound occurred at age 66 months (5.5 years). Accordingly, every rebound occurring before this age in your population should be considered and classified as early.

Finally, the “late AR” group should be renamed “moderate-to-late AR”, as a late AR is usually an AR occurring after the age of 7, and the exams don’t go after 6 years of age in your population.

3/ During the 2nd part of the introduction, the authors insisted on SGA children but the authors do not have the information regarding gestational age of their participants. In line with my first comment, please revise this part.

4/ In Table 1, it is stated than 100% of the participants reached the AR by 69 months of age, whereas, in the methods section, it is stated that participants classified as “late AR” where those who had “no AR until 69 months of age”. When looking at the BMI curve, it seems that there is no AR for participants in the “Late AR group”. Please, clarify this point. If all of the participants in this group did not reach the AR by 69 months, you cannot present in Table 1 than 100% of the participants experienced the AR by 69 months.

5/ Why did you not use the usual definition of rapid weight gain? (increase of more than 0.67 in weight-for-age z-score from birth to 6 months)

6/ Methods; please, elaborate regarding the questions asked/questionnaires regarding lifestyle habits performed and when they were performed, because we only discover this information in the results section.

7/ Statistics section; Please, add the modeling of the BMI percentile curves.

8/ Results, section “BMI trajectories during the first 6 years”

In this section, you compare the BMI values at each time points between girls and boys, and these values were different at each point. However, it does not necessary mean that boys and girls do have a different BMI trajectory. Please, can you specify how you define “BMI trajectory” and how you statistically found that those trajectories are different between boys and girls?

9/ Still in the “BMI trajectories during the first 6 years” results section; you say that 10,209 subjects were in the EAR group, 2,697 in the moderate AR group and 3,301 in the “late AR group”. So there is, in total, 16,207 subjects. However, in table 2, where you show participants with BMI values at age 57 months (exam 6) you only have 11,106 subjects left, but your Figure 1 (flow chart) shows that, at Exam 6, you have 17,590 participants. Please, explain why you lost so many participants (which constitute almost the half of your EAR group).

10/ Results, section “The timing of adiposity rebound”.

I do not understand the following sentence “The median and mode of the timing of AR was 45 months (Exam-V) and 33 months (Exam IV) of age, respectively, in all subjects”. Please, clarify or correct.

11/ Table 2: Instead of investigating the OR for adiposity rebound depending on body weight status, I suggest you rather investigate the OR for underweight, overweight or obesity depending on the timing of the AR.. as the AR is known as being a risk factor of obesity. Then, you could adjust your regression on various factors such as breastfeeding, BMI of the parents, screen use, sugar-sweetened beverage intake at 21 months, exercise habits etc..

12/ Table 3: Instead of showing percentage according to the dependent variable (EAR, moderate or late AR), you should show them according to the independent variable, between the different groups (e.g Among all breastfeed children, please show the proportion of them being in the EAR, the Moderate and Late AR groups).

13/ The manuscript should be revised by an English-native person, especially the results section which is unclear and clumsy.

14/ Just a thought: among your participants with “EAR” (45 months), about 25% will be overweight/obese by 5 years old. It also means than 75% of them should not experience weight issue (at least at 5 years of age). This maybe suggests that the classic way to define the adiposity rebound is not discriminant enough and calls for a new approach/definition in order to target children at risk for later obesity.

6. PLOS authors have the option to publish the peer review history of their article (what does this mean?). If published, this will include your full peer review and any attached files.

Reviewer #1: No

---

## [Author Response · Author response to Decision Letter 0]

14 Sep 2020

We appreciate the reviewer’s comments, which have helped us to significantly improve our manuscript. We attached Response Sheet as a file.

---

## [Decision Letter · Decision Letter 1]

1 Oct 2020

Body mass index trajectories and adiposity rebound during the first 6 years in Korean children: based on the National Health Information Database, 2008-2015

PONE-D-20-11438R1

Dear Dr. Kang,

We’re pleased to inform you that your manuscript has been judged scientifically suitable for publication and will be formally accepted for publication once it meets all outstanding technical requirements.

Kind regards,

Jean-Philippe Chaput, Ph.D.

Academic Editor

PLOS ONE

Additional Editor Comments (optional):

The reviewer felt that the comments have been adequately addressed in a revised version of the paper. Therefore, the paper is now acceptable for publication in PLOS ONE.

Reviewers' comments:

Reviewer's Responses to Questions

**Comments to the Author**

1. If the authors have adequately addressed your comments raised in a previous round of review and you feel that this manuscript is now acceptable for publication, you may indicate that here to bypass the “Comments to the Author” section, enter your conflict of interest statement in the “Confidential to Editor” section, and submit your "Accept" recommendation.

Reviewer #1: All comments have been addressed

2. Is the manuscript technically sound, and do the data support the conclusions?

Reviewer #1: Yes

3. Has the statistical analysis been performed appropriately and rigorously? 

Reviewer #1: Yes

4. Have the authors made all data underlying the findings in their manuscript fully available?

Reviewer #1: No

5. Is the manuscript presented in an intelligible fashion and written in standard English?

Reviewer #1: Yes

6. Review Comments to the Author

Reviewer #1: (No Response)

7. PLOS authors have the option to publish the peer review history of their article (what does this mean?). If published, this will include your full peer review and any attached files.

Reviewer #1: No

---

## [Editor Report · Acceptance letter]

21 Oct 2020

PONE-D-20-11438R1 

Body mass index trajectories and adiposity rebound during the first 6 years in Korean children: based on the National Health Information Database, 2008-2015 

Dear Dr. Kang:

I'm pleased to inform you that your manuscript has been deemed suitable for publication in PLOS ONE. Congratulations! Your manuscript is now with our production department. 

Kind regards, 

on behalf of

Dr. Jean-Philippe Chaput 

Academic Editor

PLOS ONE